# Adjustment of the Life Cycle Inventory in Life Cycle Assessment for the Flexible Integration into Energy Systems Analysis

**Thomas Betten [1,*], Shivenes Shammugam [2] and Roberta Graf [3]**

[1] Institute for Acoustics and Building Physics, University of Stuttgart, 70563 Stuttgart, Germany

[2] Fraunhofer Institute for Solar Energy Systems (ISE), 79110 Freiburg, Germany; shivenes.shammugam@ise.fraunhofer.de

[3] Fraunhofer Institute for Building Physics (IBP), 70563 Stuttgart, Germany; roberta.graf@ibp.fraunhofer.de

**\*** Correspondence: thomas.betten@iabp.uni-stuttgart.de; Tel.: +49-711-9703168

**Abstract:** With an increasing share of renewable energy technologies in our energy systems, the integration of not only direct emission (from the use phase), but also the total life cycle emissions (including emissions during resource extraction, production, etc.) becomes more important in order to draw meaningful conclusions from Energy Systems Analysis (ESA). While the benefit of integrating Life Cycle Assessment (LCA) into ESA is acknowledged, methodologically sound integration lacks resonance in practice, partly because the dimension of the implications is not yet fully understood. This study proposes an easy-to-implement procedure for the integration of LCA results in ESA based on existing theoretical approaches. The need for a methodologically sound integration, including the avoidance of double counting of emissions, is demonstrated on the use case of Passivated Emitter and Rear Cell photovoltaic technology. The difference in Global Warming Potential of 19% between direct and LCA based emissions shows the significance for the integration of the total emissions into energy systems analysis and the potential double counting of 75% of the life cycle emissions for the use case supports the need for avoidance of double counting.

**Keywords:** life cycle assessment; energy system analysis; energy system modelling; decarbonisation; double counting; direct emissions; resource use

## 1. Introduction

In the future, energy systems will need to be predominantly shaped by renewable energy sources [1] if the targets of the 2030 Agenda for Sustainable Development including the Sustainable Development Goals are to be met [2]. The necessary effective energy system transformation is a process that requires deep understanding of the energy system including all relevant technologies; infrastructure; as well as the political, economic, and ecological framework. Energy system analysis (ESA) offers the possibility to evaluate the interaction of these aspects and their impact on the development of the energy system. The holistic perspective of ESA makes it indispensable for scientifically sound and reliable decision-making process in politics. Energy system optimisation models are tools that facilitate ESA by mathematically modelling and optimising the entire expansion of the energy system and the operation of power plants under provided conditions and restrictions. Some of the available energy system models include—but are not limited to—PRIMES [3], Enertile [4] and REMod [5]. Although the different modelling approaches and the key assumptions might vary, these models share the same objective in minimising total system costs in order to obtain the necessary investments and operational strategies to achieve a targeted reduction

of the greenhouse gas (GHG) emission in a given time period [6]. However, cost optimisation does not necessarily reflect the real pathway of the most suitable energy transition [7].

In energy system models, GHG emission reduction can be achieved by replacing conventional power plants with renewable energy technologies subsequently over a certain period to time (other potential options, among others, include carbon removal technologies). The operational emissions from conventional power plants are also described as direct emission and mainly derive from the burning of fossil fuels. Regarding the direct emissions, renewable energy technologies are generally considered to be carbon-neutral with no direct emissions occurring during the use phase (e.g., electricity generation process). The total life cycle emissions (sometimes also incompletely referred to as indirect emissions) on the other hand refer to the total emissions emitted during the entire life cycle of a power plant, which includes the mining of relevant materials, manufacturing processes, transportation, operation, end-of-life treatment as well as the use phase emissions during operation. In ESA, all mentioned emissions are generally attributed to the country where the emissions take place (as direct emissions). This reflects a production-based perspective. The failure to allocate life cycle emission to the consuming entity is considered to be a drawback in current energy system models [8,9]. One of the consequences of this is the occurrence of carbon leakage, which outsources emissions and the corresponding environmental impact beyond the system boundaries of the assessed energy system. Ultimately, this would undermine the effectiveness of climate regulations and subvert the competitiveness of national industries [10]. For example, by the end of 2019, almost 48.6 GWp of photovoltaics (PV) was cumulatively installed in Germany, with 2.9 GWp and 3.9 GWp installed in 2018 and 2019, respectively [11]. According to the German Environment Agency UBA, 187.3 million tons of $CO_2$-equiv. was avoided in 2018, of which 15.3% was attributed to PV [12]. However, this does not give the whole picture as the emissions to manufacture the renewable energy technologies are not considered. For a PV system, the life cycle emissions can reach up to 0.08 kg $CO_2$-equiv./kWh [13], whereas for wind turbines, the life cycle emission can reach up to 0.032 kg $CO_2$-equiv./kWh [14] considering the whole life cycle. The adoption of a consumption-based perspective would be beneficial for ESA and its claim to policy consultation, as it is a fairer way to allocate emissions to the different countries or regions.

The reason as to why energy system models in the past were focused on direct emissions is on the one hand the difficulty of obtaining total life cycle emissions, and, on the other hand, the questions of whether and how emissions attributable to technology and fuel imports should be credited to the nationally determined contributions under the Paris Agreement (cf. [15]) remain unclear. Due to these reasons, the indirect emissions of renewable energy technologies are attributed to the country of origin, which often leads to a concentration of allocated emissions in small number of countries. In Germany, for example, almost all PV modules are imported from Asian countries such as China, Taiwan or Malaysia [16,17]. In Germany, the direct emission from 1 kWh of electricity at the end user in 2017 was 0.485 kg of $CO_2$-equiv. [18]. The corresponding value for the life cycle emissions is 0.575 kg of $CO_2$-equiv. (see in [19], year 2016, grid mix, Environmental Footprint 3.0). This difference of 19% will even increase in the future with an increased share of renewable energies in the electricity grid, as the share of direct emissions from conventional energy technologies will decrease [20]. Yet, a major problem of the integration of life cycle emissions in ESA is the feasibility because of the time-consuming Life Cycle Assessment (LCA) modelling process and the potential of double counting emissions within the two methodologies [21].

The main purpose of this study is to investigate the significance of avoided double counting in attributional LCA for the integration in ESA and provide an easy to use and modular approach to face the requirements for its avoidance. The underlying research questions are as follows. What is the dimension of the potential double counting and how sensitive are the LCA results towards the choice of system boundaries related to the double counting? The work focuses on the feasibility of the Life Cycle Assessment while ensuring the avoidance of double counting. In this study, based on the determined requirements necessary for integration into ESA, the system boundaries of LCA are adjusted. The approach takes existing theoretical methods and transfers those into a flexible and easy-to-use approach within the field of LCA by adjusting the Life Cycle Inventory. While a

demonstration for GHG emissions, expressed as the Global Warming Potential (GWP), is conducted in the study, future integration of further impact categories (e.g., resource use indicators) and exchanges with the ecosphere on inventory level (e.g., copper) into ESA should be feasible with the same approach. The process and the impact on the results are demonstrated on the use case of Passivated Emitter and Rear Cell (PERC) photovoltaic technology on a technology level.

## 2. Materials and Methods

The methodological approach of the integration of LCA into ESA is based on the understanding of ESA, which is described in Section 2.1.1, and the concept of LCA, which is described in Section 2.1.2. These two sections are part of a state-of-the-art review, which is accompanied by a short status of the recent research on the integration of LCA into ESA in Section 2.1.3. The description of the state of the art focuses not only on a brief overview of methods but also on the details that explain necessary adjustments for the integration, e.g., system boundaries. The applied approach for this study is presented in Section 2.2.

### 2.1. State-Of-The-Art

#### 2.1.1. Energy Systems Analysis

The basic working principle of energy system analysis is that the energy provision of an investigated region has to match the demand at every time step, subject to a set of constraints such as the grid capacity, power plant availability or the weather conditions. However, the basic methodology of the energy system models used to perform such an analysis may vary. For example, there are energy system simulation models that require relatively little computing time [22]. Some examples of simulation models are found in [23,24], which use a bottom-up approach that includes detailed modelling of the technologies in an energy system. Other models such as those in [25] use the agent-based simulation approach, where relevant actors in an energy system such as the network operators or municipal utilities are individually modelled with specific properties. Simulation models are usually applied to capture technological behaviour as realistically as possible. Instead of finding out the most cost-effective solution, they are often used to investigate responses to policy shocks [26].

In addition to that, models for optimising the energy system exist. Contrary to simulation models, optimisation models optimise power plant expansions and the unit commitments so that the demand can be fulfilled at the least cost while ensuring constraints such as a limit for $CO_2$ emissions. Optimisation models represent the majority of the models currently used in research [27]. Although optimisation models do not necessarily need to reflect the actual investment behaviour or unit commitments of power plants in reality, the results are crucial as they represent the full economic savings potential that can be achieved with better planning and intelligent management systems. Most energy system optimisation models, such as those in [28–32], use a linear programming approach, which simplifies the complexity of the energy system in linear mathematical equations. However, optimisation models can also be nonlinear, such as that in [33], where parts of the constraints of the models are represented in nonlinear equations. The disadvantage of this approach is however the increase in computing time [22].

Besides that, there are also equilibrium models [3,34], which describe the energy system as part of the national economy. These models often use the utility function to investigate the relationship between the energy sector and other sectors of the economy. These models usually do not have a high technological resolution as their aim is to analyse the cost-effectiveness of climate policy measure while considering the entire economic system. A subcategory of the equilibrium models is the partial equilibrium models [35,36], in which only the energy sector is specifically modelled while assuming that the conditions in the rest of the economy remain unchanged.

Apart from the equilibrium models, which do not consider individual technologies, all other aforementioned energy system models have in common that each energy technology is assigned a specific $CO_2$ emission factor, usually expressed in the amount of $CO_2$ emissions per unit of generated

electricity (kg $CO_2$/kWh$_{el}$). For instance, a lignite power plant with an emission factor of 0.403 kg $CO_2$/kWh$_{lignite}$ and an average efficiency of 34% leads to an assessment factor of 1.18 kg $CO_2$/kWh$_{el}$, whereas for a hard coal power plant with an efficiency of 38% and an emission factor of 0.34 kg $CO_2$/kWh$_{hardcoal}$, the specific emission factor amounts to 0.89 kg $CO_2$/kWh$_{el}$ [37]. New technologies, such as PV, wind turbines or li-ion batteries, have virtually zero emissions during the use phase. Based on the total electricity generation of each technology, the total $CO_2$ emission from the entire energy system can be calculated. If the model optimises or simulates the system with limited $CO_2$ budget, then clean technologies will be favoured while operation and expansion of conventional power plants are gradually supressed. In this way, the future energy system of less $CO_2$ emissions can be established.

In order to reduce the $CO_2$ emissions on the national level, it is only feasible to consider the emissions within the system boundary of the model, which in most cases is the modelled country [37,38] or region such as the EU [4]. A recent energy system study for Germany estimated that up to 160 GW of cumulatively installed capacity of PV is required in 2050 in order to fulfil the national climate goal of GHG-reduction [39]. However, as the majority of modules are imported from China, the emission responsible for the manufacturing of PV modules is attributed to China, and not Germany. This principle is known as the polluter pays principle, which dictates that whoever is responsible for the production of a good, should bear the pollution that result from it. The polluter pays principle contends that firms and, in turn, countries should be charged for the full costs to society of their current pollution [39]. Therefore, from the perspective of energy system analysis, the emissions due to manufacturing of PV modules that are to be exported to Germany are to be considered in the energy system optimisation of China. To allocate the emissions more fairly among the responsible entities, a consumer pays principle would be beneficial. In this way, countries could on the one hand be charged for the full costs to society and on the other hand, assess where the most efficient levers regarding their actions are.

### 2.1.2. The Fundamentals of Life Cycle Assessment

In this section, the methodology of attributional LCA is described, with a focus on the aspect of life cycle thinking. The methodology of LCA is used to assess the environmental impacts of a product or service over its entire life cycle. Once the goal and scope have been defined, data is collected in order to analyse the inventory of all relevant material and energy flows associated with the product life cycle. Then, all flows are characterised according to its impact on different environmental categories. The best known and mostly used impact category is Global Warming Potential, aggregating the impacts of each inventory flow in kg of $CO_2$-equivalents, expressing the total impacts of the product on climate change caused by greenhouse gases [40].

The perspective of life cycle thinking enables LCA to cover the total environmental impacts associated with the product (e.g., energy system technology). The system boundaries focus on the product life cycle and are mostly indifferent to political entities. Geographical differences can be accounted for in the background data [41]. Thus, the methodology of LCA follows a consumer pays principle as it allows producing consumption-based footprints [41,42]. While LCA is standardised in ISO 14,040 and 140,444 [40,43], there is no such thing as a standard LCA as the goal and scope is always dependent on the requirements and the purpose of the study. The term standard LCA will here be used to describe LCA studies that are not adopted to be used in ESA and thus covering the whole life cycle. The main advantages and reasons why LCA is established as an important sustainability assessment tool are the capability to quantify burden shifting and to identify trade-offs by covering a wide range of impact categories [44].

2.1.3. The Integration of LCA-Based Indicators into Energy Systems Analysis

Energy systems analysis, in general, focuses on the assessment of the energy system for a confined geographical area. On the contrary, LCA focuses on the assessment of environmental impacts over the whole life cycle of a product. This difference and commonalities in the system boundaries are visualised in Figure 1. In particular, the direct emissions from the involved energy technologies can be covered by both methodologies [21].

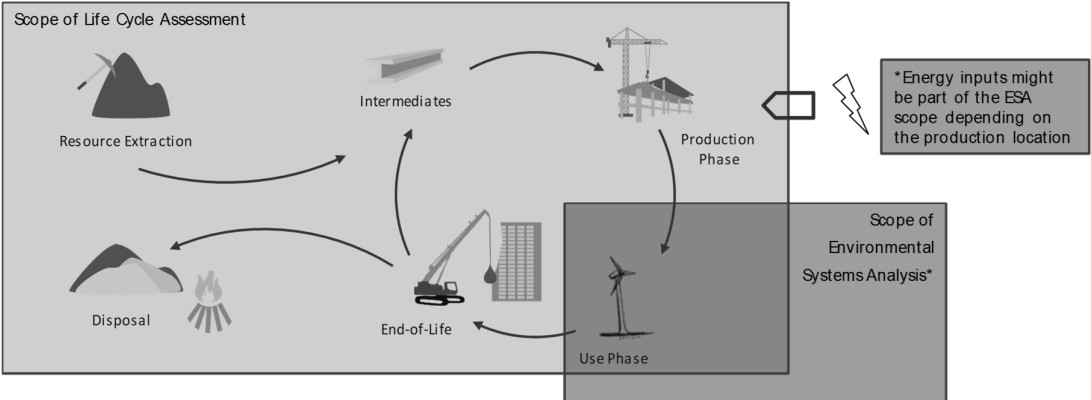

**Figure 1.** Graphical representation of the difference in the system boundaries for Life Cycle Assessment (LCA) and Energy Systems Analysis (ESA). LCA includes the environmental impacts from the whole life cycle, while ESA focuses on direct emissions from the use phase of, in this case an energy technology (energy inputs to other life cycle phases might be included depending on the production location).

When integrating LCA into ESA several issues need to be considered which are of common interest for the sound integration of the two methodologies. Blanco et al. [44] lists the following common issues when combining LCA with ESA.

- Double counting,
- Imports and exports,
- Spatial differentiation,
- Temporal differentiation,
- Biomass emissions,
- Multifunctional processes, and
- Future performance of technologies.

While some of the issues are only relevant on a system level (e.g., imports and exports) and others are only relevant for specific environmental indicators (e.g., spatial differentiation), issues like double counting are always relevant for the integration. This study and thus the conducted literature review focus on the double counting and the demonstration of its impacts. This focus was chosen as it is considered as key for the derivation of tangible actions from the ESA.

Double counting describes the accounting of direct emissions within the life cycle of an energy technology in the LCA and at the same time on system level in the ESA (cf. Figure 1). After the integration of both methodologies, specific flows are counted twice. The prerequisite for the double counting is that the production of the energy technology takes place in the country that is addressed in the ESA. As an example, the electricity required for the manufacturing of the technology is included in the total electricity demand, for which the energy system model optimally dispatches power plants. Therefore, the $CO_2$ emissions for the production of the technology are partly considered in the ESA. Double counting of the emission occurs when the entire life cycle of a technology is modelled in the LCA without adjusting the system boundaries according to the assumptions of the ESA model. While the integration of LCA into ESA is widely practiced [8,21,44–48] and its benefits in term of accuracy by taking more emissions into consideration are recognised

[21,44,47], the integration is not always done very meticulously as double counting of emissions is often not avoided and formalisation is needed [8].

The integration of LCA in ESA including the avoidance of double counting is described in detail by Volkart [21] and Volkart et al. [49] with the proposed steps being listed in the following.

1.  Matching energy system technologies with their corresponding Life Cycle Inventory (LCI) datasets
2.  Subdividing LCI datasets according to the life cycle phases
3.  Constructing a background LCI database without the energy system of the considered region(s)
4.  Calculating the cumulative LCI and conducting Life Cycle Impact Assessment (LCIA).

The method proposed by Volkart is comprehensive for the theoretical approach and covers the application for matrix calculation in process LCA. Yet, the issue of double counting is addressed very seldom in studies including life cycle emissions in ESA [21]. Existing examples for the integration of LCA indicators based on emissions in ESA can be found in (I) García-Gusano et al. [45], where LCA based emissions are integrated into a TIMES model [50]; in (II) Loughlin et al. [46], where LCA indicators from the GREET model are integrated into ESA; and in (III) Naegler et al. [47], in which LCA indicators are used to overcome the bias of optimisation by cost parameters only. Additionally, Rauner and Budzinski [48] propose a systematic integration of LCA in ESA. In addition to environmental and economic indicators, Eckle et al. [51] also include social factors (e.g., the risk from severe accidents) in the assessment of the energy system. While integrating LCA indicators into ESA, the studies mentioned above do not address the problem of double counting. The dimension of the potential overestimation remains unclear and problematic if the results are to be used in decision making processes. Naegler et al. [47] acknowledge the prevalent problem of double counting, and Astudillo et al. [8] calls for a solution for the double counting problem and a formalised common language between LCA and ESA to increase the value of the integration. Volkart [21] and Blanco et al. [44] address the problem. Volkart applies her methodology (described above) to global energy scenarios, whereas Blanco implemented his proposed approach to power-to-methane production in the European Union. Yet, no easy-to-use guideline or process is outlined in the studies either.

The problem of double counting is in particularly prevalent in hybrid Life Cycle Assessment approaches and discussed thoroughly in the scientific community. Strømman et al. [52] and Agez et al. [53] provide a profound compilation of the available computational options. While the underlying problem is similar, the approaches cannot be transferred directly to the integration of LCA indicators into ESA as the approaches target the environmental extended input–output analysis and not the process LCA. The process of integrating the results of ESA into Life Cycle Assessments is more widespread than the opposite. It is done mostly in the form of using the energy composition of a national grid to compose the electricity background data in LCA. Two examples are conducted in Navas-Anguita et al. [54], where a LCA of induced change in the environmental systems analysis through the adoption of electric vehicles in Spain is conducted, and in Pichlmaier et al. [55], where the dynamisation of LCA by the integration of ESA dynamically is described. Pichlmaier et al. declare the plan to integrate LCA indicators in their ESA models. A summary of integration cases of ESA in LCA is given in Blanco et al. [44].

## 2.2. Adjustment of the Life Cycle Inventory for Flexible Integration into Environmental Systems Analysis

The proposed approach in this paper is based on the theoretical method developed by Volkart [21] and focuses on the feasible and flexible integration of LCA based indicators into various models within ESA. In doing so, the approach aims to counteract the slow uptake of the integration of LCA based indicators in ESA, considering the avoided double counting. The standard approach to LCA is adjusted to fit the system boundaries of ESA. Figure 2 illustrates the approach giving credit to the fact that LCA is integrated into ESA. The advantage of this approach is that it only adapts the system boundary of LCA, so that the existing ESA models can be used without any restrictions.

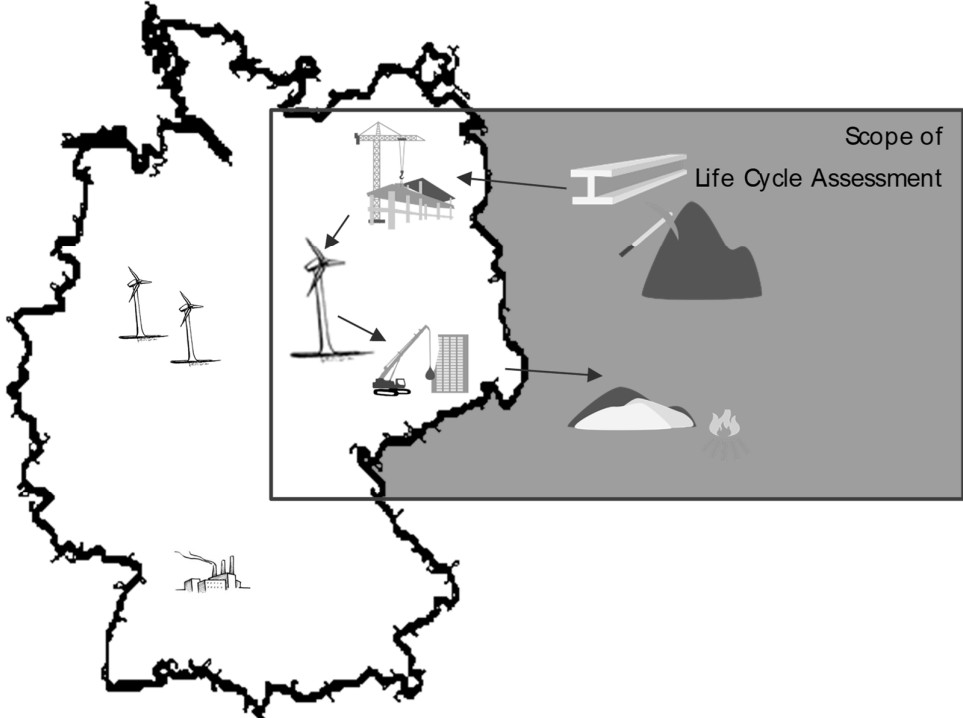

**Figure 2.** Graphical representation of the necessary adjustment in LCA for an integration into ESA with the geographical scope of Germany. Grey parts of the life cycle are included in the assessment; parts covered by the systems under investigation in the ESA are cut off to avoid double counting as the emissions are already covered by the ESA model.

In order to ensure the easy and flexible use of the approach, the following main requirements towards the process design were considered in the development process.

- The adoption of a consumption-based approach in ESA must be enabled (by extending the emission scope to indirect emissions from imported products used in the energy system).
- Double counting of direct emissions should be eliminated by alignment of the system boundaries in LCA to increase the accuracy of the results.
- No changes in the assumptions and methodology of ESA should be necessary.
- Modelling efforts in LCA should be kept at a feasible level to increase uptake.
- Discrete-time evaluation of LCA results by life cycle phases to obtain a temporal resolution of the ESA in order to account for the impact at the time of commissioning of the energy technology.

Accordingly, Volkart's method [21] was adapted as deemed suitable for operation. The adapted approach used in this study is described below, whereas the steps described in italics are additional steps developed by the authors within the scope of this paper:

1. Identification of technology surrogate models

In order to keep the modelling efforts on a feasible level, surrogate technologies are defined for the different energy technologies in ESA. For the use case (see Section 3.1), a PERC photovoltaic system is used as the surrogate technology to cover the ESA component of photovoltaic power. In this step each specific technology addressed in the ESA is linked with an LCA model that provides associated life cycle impacts.

2. Matching energy system technologies with their corresponding Life Cycle Inventory (LCI) datasets

The model is set up according to the standard LCA methodology (see Section 2.1.2). Restrictions coming from step 3, 4 or 5 can already be addressed in this step or subsequently. This allows for a flexible process based on the status of the available information and existing LCA models. Laurent et al. [56] describe an approach to modelling energy systems in LCA that can be used in the step.

3.  Subdividing LCI datasets according to the life cycle phases

The LCI model is subdivided for the different life cycle phases. This step is necessary as certain phases are covered by ESA models, such as the as direct emissions, of conventional power plants. As for renewable energy technologies, the time of the occurrence of the environmental impacts is key for some categories and its evaluation within ESA (e.g., resource use). This step does not eliminate the potential double counting of emissions in the production phase of the energy technology.

4.  Constructing a background LCI database without the energy system of the considered region(s)

The renunciation of the holistic consideration of all life cycle impacts is necessary to avoid double counting. Energy inputs that are covered in the LCA model and also in the ESA model need to be cut off. The cut-off in LCA is conducted in order to ensure a smooth integration into existing ESA models. For example, if the assessed energy system boundary is Germany, all production energy going into the energy technology is already accounted for either in the year of commissioning or earlier.

5.  Dynamisation of the background model

The dynamisation of the background model, especially for energy inputs, is necessary to account for the potential decarbonisation of the energy system that is used to produce the energy technology. This step is vital to avoid a lag of time representativity that is induced by the use of non-conforming LCA models. The avoidance of double counting regarding production emissions is also necessary for the background model data.

6.  Calculating the cumulative LCI and conducting Life Cycle Impact Assessment (LCIA)

The calculation of the environmental impacts is optional but can help to broaden the scope of the conducted analysis. This step does not differ from the standard LCA approach described in Section 2.1.2.

The used approached is shown as a schematic flow chart in Figure 3. The selection of the technology surrogates is dependent on the used ESA model, but can be the same for different ESA studies.

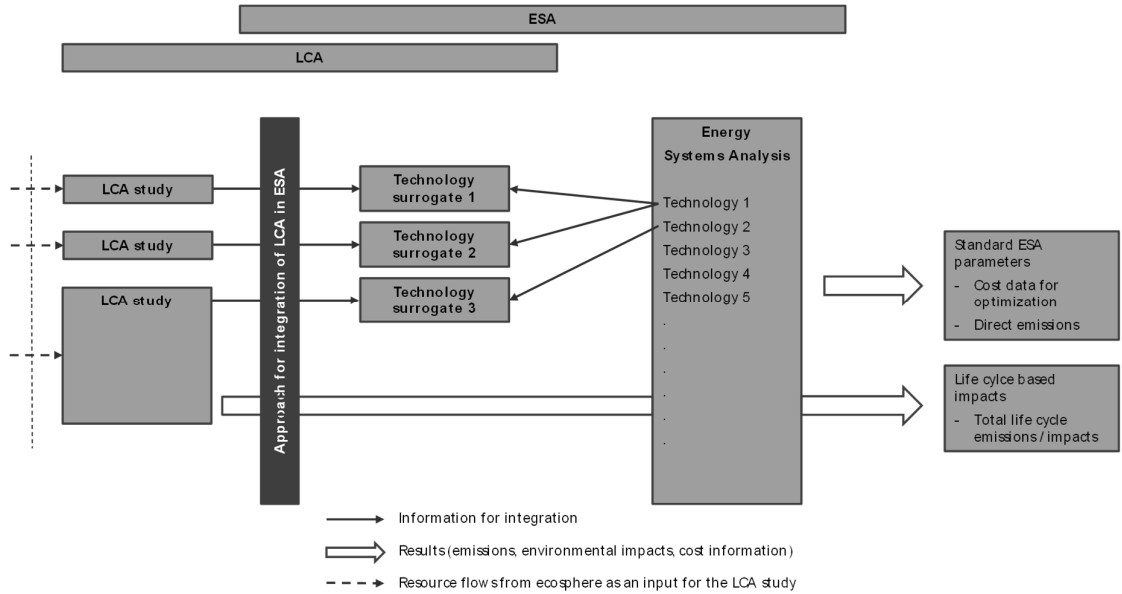

**Figure 3.** Schematic flow chart of the adjustment of LCA models for the adoption to ESA.

The focus of the proposed approach is the strict link to process LCA and the easy-to-use application process. Table 1 shows an overview of the different steps and a short description of the process (also described in detail above). In addition, it links the different steps to a purpose based on the requirements formulated above. The following case study is meant to illustrate the relevance of the proposed procedure.

**Table 1.** Short description of the alignment of the Life Cycle Inventory (LCI) dataset for the integration into ESA.

| Step | Details | Purpose |
|---|---|---|
| 1. Surrogate Models | Identification of suitable technologies to represent the technology class in ESA | Ensuring feasibility of modelling |
| 2. LCI Matching | Modelling of the technology surrogate including changes over time | Consumption-based approach |
| 3. Subdivision | Enabling the time discrete assessment of environmental impacts | Discrete time evaluation |
| 4. Double Counting | Avoidance of overestimation of impacts from life cycle | Increasing consistency |
| 5. Dynamisation | Dynamisation of background datasets | Increasing accuracy |
| 6. LCIA Calculation (Optional) | - | Broadening the assessed impacts |

## 3. The Use Case of PERC Solar Technology—Description and Results

The assessed use case is intended to give an insight into the importance and significance of the consideration of the right system boundaries when integrating the results of an LCA study in an ESA with the geographical scope of Germany. In Section 3.1, the use case is described. First, the technological details are elaborated, before the data for the life cycle inventory are described. Afterwards, the assessed scenarios are introduced. Section 3.2 presents the results of the LCA study.

### 3.1. Description of the Use Case

#### 3.1.1. Technology Description

The global PV market is currently dominated by Silicon (Si)-based solar cells. Among various cell concepts, the global market by the end of 2018 was dominated by Aluminium-Back Surface Contact (Al-BSF) cells, with 60% of the market share, followed by a share of 35% for Passivated Emitter and Rear Cells (PERC). However, based on the prediction of the International Technology Roadmap for Photovoltaic (ITRPV) [57], Al-BSF is expected to be phased out by 2025 as PERC will take over as the market leader. This is consistent with the current trend of increasing market share of PERC solar cells in the industry. One of the main advantages of PERC solar cells is that they allow for higher cell efficiency at relatively low cost. While the efficiency of Al-BSF is limited to ~20%, PERC offers reduced rear-surface recombination, improved internal rear reflectivity and higher infrared light absorption, with a cell efficiency record lying currently at 25% [58,59]. As the cell topology is quite similar to Al-BSF, the industrial equipment and processes can still be used for PERC solar cells, with only minor changes required [60,61]. Therefore, a typical commercial PERC is considered for the use case of the proposed approached in this paper. The cross-section of a PERC solar cell is illustrated in Figure 4. The technical functionality of Al-BSF and PERC solar cells are detailed in Hargett et al. [62].

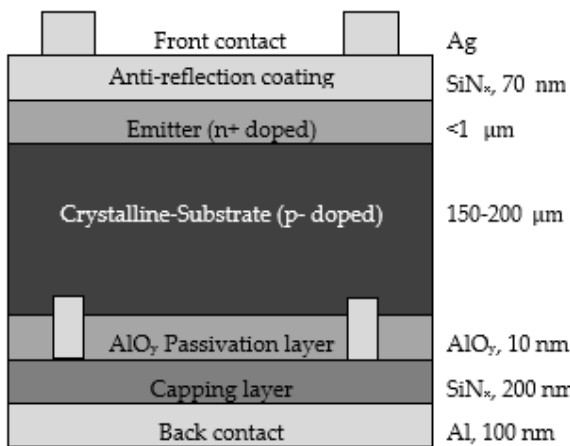

**Figure 4.** Schematic representation of the cross-section of a Passivated Emitter and Rear Cells (PERC) solar cell including a description of the different layers and its thickness.

### 3.1.2. Life Cycle Inventory

The life cycle inventory of the LCA model for the reference year of 2018 is based on the work by Frischknecht et al. [63] under the Photovoltaic Power Systems Programme by the International Energy Agency. If not stated otherwise, the inventory was adopted from this work and based on the Crystalline–Si PV technology. Adjustments for the PERC technological and new developments in production technologies are described in the following.

The specific material content of the PERC modules is assumed to reduce linearly with the development of the module efficiency, which is expected to increase from 18.5% in 2020 to 24% in 2050. Apart from that, several other material efficiency measures are also taken into consideration. For example, the silicon consumption is assumed to reduce through thinner silicon wafers, reduced sawing losses and lower breakage rates. Currently, a thickness of 175 μm is preferred for the crystalline substrate of PERC-Si wafers. N-type solar cells offer the potential to reduce the wafer thickness faster than p-type cells. Assuming that n-type wafers gain more market share in the next few years, an average wafer thickness of ~120 μm can be expected to be standard in 2050 [57].

One of the most important cost drivers in module manufacturing is the use of silver for contact formation. Therefore, continuous efforts have been carried out to effectively reduce the use of silver without compromising the efficiency. The silver content in the PERC solar cell is expected to decrease as a result of new developments in metallisation pastes, screen printing processes such as dual printing as well as alternative metallisation designs such as thinner fingers and busbarless configuration [57]. Another major change that can be expected in PV modules is the use of lead-free soldering in 2050, via the substitution of lead with bismuth [64]. Besides that, PV module mounting frames currently consist mainly of aluminium. Frameless modules, on the other hand, offer aesthetic advantages as well as easy installation and a longer service life by reducing potential-induced degradation and eliminating frame corrosion problems. Assuming that the commercial range of suitable mounting systems continues to develop, frameless design should therefore become the market leader in the long term. Considering all aforementioned material efficiency measures, the specific material demand for a PERC-based PV module is extrapolated until 2050 and is shown in Table 2 for the year 2018 and 2050.

**Table 2.** Development of the specific material composition of PERC modules under consideration of material efficiency measures and improvement of module efficiency.

|  | 2018 | 2050 |
|---|---|---|
| **Module [kg/MW]** | | |
| Al—Aluminium | 184 | 130 |
| Ag—Silver | 21 | 7 |

| | | |
|---|---|---|
| Cu—Copper | 946 | 670 |
| Pb—Lead | 62 | 0 |
| Bi—Bismuth | 0 | 47 |
| Si—Silicon | 1441 | 826 |
| Sn—Tin | 59 | 35 |
| Glass | 44,881 | 31,791 |
| Ethylenvinylacetate | 5058 | 3583 |
| Polyphenylenether, polystyrol | 530 | 375 |
| Polyethyleneterephthalate | 2333 | 1652 |
| Polyvinylfluoride | 1063 | 753 |
| Acrylic foam | 218 | 154 |
| **Frame [kg/MW]** | | |
| Al—Aluminium | 8768 | 0 |

3.1.3. Assessed Scenarios

In order to examine the significance of the double counting, four scenarios for the assessment of the PERC technology were developed. For the purpose of this study, the LCA was reduced to the assessment of the production phase. The scope of the ESA for integration is the German national energy grid. The assessed scenarios are listed in the following.

- *Standard* LCA: In this scenario, the system boundaries of the LCA study remain unchanged compared to conventional LCA and the idea of life cycle thinking. The model accounts for all impacts along the life cycle. Thus, the LCA model is not adjusted to avoid double counting when being integrated into ESA. The reference year is 2018 and the PV modules are produced in China, with all other parts being produced in Germany. This scenario represents realistic production assumptions (see Section 3.1.1).
- Production in *China* (including avoided double counting): This scenario is based on the status quo (see Section 3.1.1.), and thus module production in China is assessed including the energy inputs, but the other technology components are produced in Germany and thus without energy inputs to avoid double counting of the occurring emissions. The reference year is 2018.
- Production in *Germany* (including avoided double counting): This scenario is based on the hypothetical scenario where PV production is retransferred to Germany. All production, including the modules, takes place in Germany and thus the energy inputs are not part of the LCA model (Note: This includes the silicon production). The reference year is 2018.
- Production in *China 2050* (including avoided double counting and including temporal dimension with the adjustment for future developments): Based on the scenario *China*, a future production in 2050 is assumed. Changes in the electricity mix are modelled according to the GaBi database [19]. The reference year is 2050. Production sites remain unchanged compared to the scenario *China*.
  Note: The decarbonisation of the material inputs was modelled for the inputs of steel, aluminium, copper, silicon and glass.

Figure 5 shows a schematic flow chart of the modelling approaches including the different system boundaries for the different assessed scenarios as described above. The scenario *Standard* represents the base case and is a conventional cradle-to-gate LCA of the technology. The scenario *China* represents the actual production situation: the wafer is produced in China, the other components in Germany. To address the double counting of the energy inputs, these are cut off in LCA as they are accounted for in ESA. The scenario *Germany* represents a hypothetical production in Germany. Thus, all energy inputs are cut off in the LCA as they are accounted for in ESA already. For the scenario *China 2050*, a temporal component was added. It is similar to the scenario *China*, but with adjusted energy input representing a production in 2050. Note: Only the scenario *China 2050* is different regarding the technological model assumptions. The difference between the other scenarios

lies in the choice of system boundaries for the integration into ESA. The different electricity processes used are detailed in Table A1 in the Appendix A.

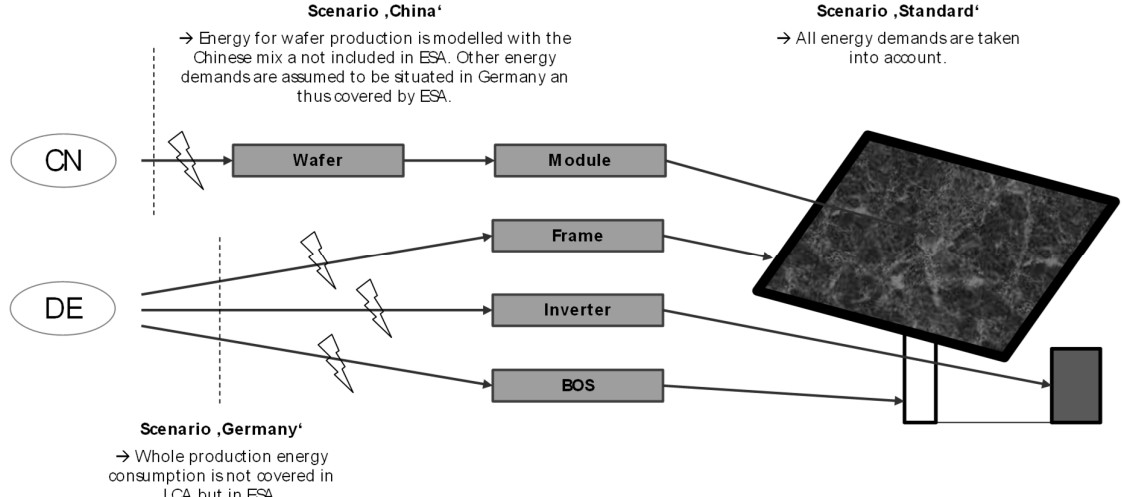

**Figure 5.** Schematic flow chart of the adjustment of LCA models in the use case for the adoption to ESA. The scenario *Standard* includes all energy inputs along the life cycle. The production groups include the module, frame, inverter and balance-of-system (BOS).

*3.2. Results*

The assessed functional unit is MWp of installed capacity. However, as the results are presented in relative numbers the reference unit is not of any particular interest. The used modelling software is GaBi by Sphera in its version 9.2.168 and the database service pack 40 [19]. For the assessment, the impact category Global Warming Potential according to the framework of the Environmental Footprint 3.0 is used [65].

Figure 6 shows the environmental impacts on the Global Warming Potential (GWP) for the different assessed scenarios. The standard LCA approach, called *Standard* scenario, was used as a reference. All inputs along the life cycle of the PERC PV technology are considered in this approach. The results for the production of the PV module in China with the avoided double counting for the energy related impacts in Germany, called *China* scenario, account for 90% of the GWP compared to the reference case. The third scenario, *Germany*, assesses the cleared environmental impacts for a production of the energy technology in Germany. The GWP accounts for 25% of the emissions compared to the reference scenario. The fourth scenario, *China 2050*, reflects a production in China in the year 2050. The energy inputs are decarbonised and so are the main material inputs (see Section 3.1.3). The emissions relevant for the GWP account for 19% compared to the reference case.

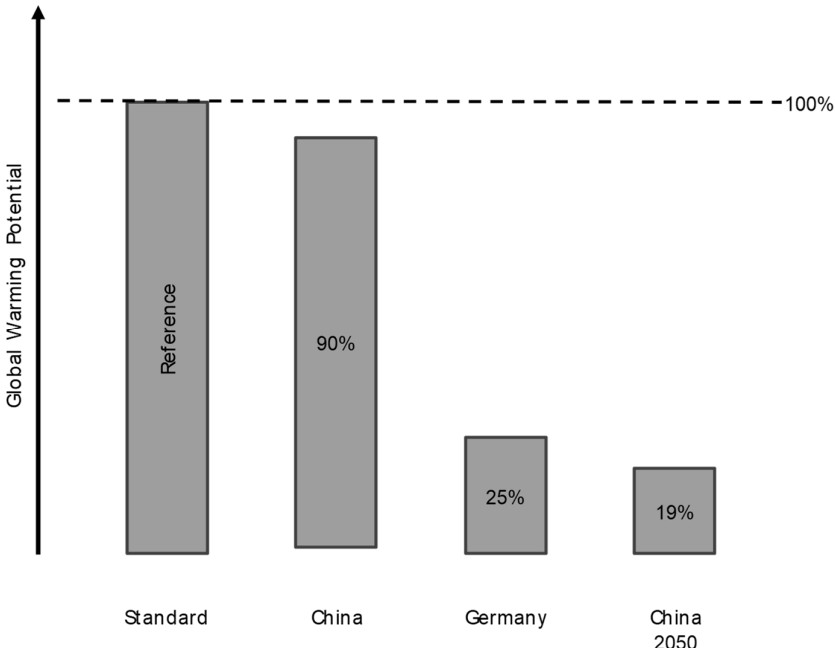

**Figure 6.** Environmental impacts (i.e., Global Warming Potential) of PERC photovoltaic (PV) for the different scenarios compared to the results of the standard LCA procedure as a reference case.

An integration into ESA is not subject of this study. The case study is meant to demonstrate the relevance of the proposed method. In this study, only GWP was assessed as it is considered to be a reasonable proxy for other environmental indicators to judge the significance of the adjustment of the system boundaries.

## 4. Discussion

With the proceeding transformation towards an energy system with an increasing share of renewable energies, the incorporation of the emissions from the whole production life cycle is getting more important. For the German electricity grid, as mentioned in the introduction, life cycle emissions are already 19% higher than the direct emissions only (reference years 2016 and 2017):

- Direct emission from 1 kWh electricity from the grid in 2017: 0.485 kg of $CO_2$-equiv. [18]
- Total life cycle emissions for 1 kWh electricity from the grid in 2016: 0.575 kg of $CO_2$-equiv. [19]

While direct emissions are a suitable indicator for an energy system dominated by fossil fuels, the discrepancy is set to increase in the future. With that trend, the correct assessment of life cycle emissions is gaining importance, while the adoption of methods avoiding double counting is not yet comprehensively implemented (see Section 2.1.3). The proposed procedure to calculate LCA-based indicators that are adjusted for double counting for the integration into ESA was tested successfully in this paper on the single case study of PERC PV. While the phase of a full integration into ESA (covering the whole scope of the ESA with all different technologies) was not realised within the scope of this study, it could be demonstrated that the double counting issue can be solved with a feasible yet accurate enough approach. The case study aimed at illustrating the sensitivity of system boundary choices to explain the relevance of the proposed adaptation.

The results show significant differences between the four considered scenarios. The differences between the scenarios *Standard*, *Germany* and *China* are derived only from the different system boundaries for the energy inputs. The significance of these differences, which can be as high as 75%, highlights the importance of correct assumptions about the production location and the scale of potential double counting when LCA indicators are integrated into ESA. The potential double counting is revealed when comparing the adjusted scenarios to the scenario *Standard*. While the

scenarios *Germany* and *China* represent different assumptions for the product life cycle, the scenario *Standard* represents the base case without considering the system boundary overlap between ESA and LCA. For the realistic production scenario *China* where the system boundary overlap was resolved, the emissions are 10% lower which represents the amount of emissions that would be double counted in *Standard* scenario. Similarly, it would be 75% of emissions that are double counted in the scenario *Germany*, with its assumption that the PV modules are produced in Germany (and thus energy inputs are excluded in the LCA). The significant difference derives from the exclusion of most production energy inputs, which in the case of China cause a major share of the impacts in particular with the high share of fossil fuels in the Chinese electricity mix.

In the scenario *China 2050*, the assumed future material and process efficiencies also influenced the result distinctly. Thus, the scenario assesses the impact of the proper technological representation for the integration. The resulting emissions without double counting and adjusted for the technological process, emphasise the need for dynamic modelling to indicate the worldwide decarbonisation and underlines the importance of step 5 of the proposed procedure for the conclusions drawn from the following ESA.

## 5. Conclusions

ESA and LCA will continue to play significant roles in the assessment of environmental impacts in their specific fields when used separately. Yet, a combination can address the issue of carbon leakage and lead to a more complete picture from ESA as not only direct emissions are considered for the calculation of the optimisation potentials. This increases the usefulness in decision-making in particular for a global scale. LCA on its own often lacks the coherence within a global context which can be provided by the integration into ESA.

The main purpose of this study is to investigate the significance of avoided double counting in LCA for the integration in ESA and provide an easy to use and modular approach to face the requirements for the avoidance. With an increased share of renewable energy technologies in the energy system, the impacts shift from the direct emissions (traditionally assessed in ESA) to the total and indirect emissions, based on LCA based indicators. Thus, the accurate calculation of these gets more and more important for the robustness of the ESA models and its implications. The applied method consists of the following steps.

- Identification of technology surrogate models
- Matching energy system technologies with their corresponding Life-Cycle Inventory (LCI) datasets
- Subdividing LCI datasets according to the life cycle phases
- Constructing a background LCI database without the energy system of the considered region(s)
- Dynamisation of the background model
- Calculating the cumulative LCI and conducting Life Cycle Impact Assessment (LCIA).

All steps were implemented in the study for the use case of PERC PV. The results show that the difference in the environmental impacts (represented by Global Warming Potential of PERC PV technology) is significant and should be addressed in the future in particular for renewable energy. The novelty of the approach is its easy implementation, in particular the calculation of the LCA results without interference with the conduction of the ESA. The approach is not tied to a single technology and can also be applied to other relevant technologies in other sectors of an energy system. One assumed cause for the slow uptake is the fact that the proportions of the potential double counting often remain unclear. The singled-out use case in this study indicates this issue. Yet, the single use case is also a major limitation and generalised conclusions cannot be drawn. Bottom-up ESA models are characterised by their technology richness [21]. Thus, for future utilisation it might be necessary to find simplified methods for the adjustment of the system boundaries, even within the approach working with technology surrogates. Yet, the avoidance of double counting is essential for a reliable statement on the energy system by ESA, especially when the models should be used for optimisation based on factors other than costs. To enable the full

potential of the integration of these two methods, some diligence and also some more research work will be needed.

In the future, the developed approach will be used to determine the resource use of the German energy transition, also called "Energiewende", within the project InteRessE [66]. For this purpose, the presented approach will be extended and consider additional methodological challenges to accurately assess the resource use of the energy transition. As resource use is time-critical due to limited and volatile market availability, the time differentiation will be one of the requirements. For LCA results on impact category level (e.g., $CO_2$–equiv. emissions as GWP) this is not particularly relevant. With the focus on resource assessment, continuing methodological adaptation is needed as the assessment of single resources and its depiction in LCA is not yet fully solved [67]. A combination of the proposed procedure with different available LCIA methods in the optional step 6 might offer additional insight. The assessed transition of the German energy system is also a transition towards a system where currently marginal technologies will play a significant role. Therefore, a fundamental question for LCA operators becomes more urgent. The applied approach of attributional LCA was selected in this paper as the goal was to allocate the relevant environmental impacts to the energy technologies used in the ESA. Yet, depending on the exact goal of the ESA, consequential LCA approaches should be considered as well (cf. [68,69]). Further exemplary fields of research could be the integration of ESA into the dynamisation of LCA-based indicators or for the opposite integration direction the internalisation of the external costs of $CO_2$ into the optimisation of ESA.

**Author Contributions:** Conceptualisation, T.B. and S.S.; methodology, T.B. and R.G.; formal analysis, T.B.; resources, T.B. and S.S.; writing—original draft preparation, T.B. and S.S.; writing—review and editing, T.B., S.S. and R.G. All authors have read and agreed to the published version of the manuscript.

**Funding:** This research was partly conducted as part of the InteRessE project (Ressourcenbedarf für die Energiewende: Interdisziplinäre Bewertung von Szenarien für die Bereitstellung von Strom und Wärme), which is funded by the German Federal Ministry for Economic Affairs and Energy under grant no. 03ET4065. The responsibility for the contents lies solely with the authors.

**Acknowledgments:** Special thanks go to Kanthaphon Herbig, Tingfeng Song and Sharon Stauffert who made this paper possible through their work on input data and/or model implementations.

**Conflicts of Interest:** The authors declare no conflicts of interest.

## Appendix A

**Table A1.** Global Warming Potential of the used electricity grid mixes for Germany and China.

| Process | Country | Year | GWP [kg $CO_2$-Equiv./kWh] | Source |
|---|---|---|---|---|
| Electricity grid mix | Germany | 2016 | 0.575 | [19] |
| Electricity grid mix | China | 2016 | 0.834 | [19] |
| Electricity grid mix (2040) (significant improvements in sustainability policy) | China | 2040 | 0.201 | [19] |

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
