# Peer review of "Adjustment of the Life Cycle Inventory in Life Cycle Assessment for the Flexible Integration into Energy Systems Analysis"

_energies, doi:10.3390/en13174437_

Round 1

Reviewer 1 Report

The paper is written on a highly actual, important and relevant topic. The assessment of environmental impact, life cycle of products, their reuse and recycling is getting rapidly more important in nowadays world. All of this is also reflecting in energy systems, generation units and all the connected devices.

More specificly the paper analyses the possibilities of how life cycle emissions can be integrated into energy systems analysis and this is shown on the example of PV cells technology in the technology level. There are comparisons done in German and Chinese examples, which is a good selection, as the latter is the biggest producer of such technologies. Moreover, it addresses straightly the ongoing discussion of the "environmental friendliness" of environmental friendly technologies.

The paper is written in good style, which is easy to read. The use of language is very good and fluent. There are no major drawbacks to be pointed out.

Author Response

Thank you very much for your feedback and commendation. Another language and spell check was conducted to further increase the quality of the manuscript.

Reviewer 2 Report

  1. General comments

The paper under revision, namely ‘Adjustment of the Life Cycle Inventory in Life cycle Assessment for the flexible Integration into Energy System Analysis’, assesses the development of a new approach to integrate Life Cycle Assessment (LCA) into Energy Systems Analysis (ESA) focused on the avoidance of carbon emission double counting.

The subject of the article is relevant for the body of literature and the paper is well articulated. However, the contribution of the study must be clarified in further details? Why is it critical for decision making? Can LCA or ESA be used separately? Overall, I suggest to publish the paper after addressing the comments outlined in Section 2 (Specific comments).

  1. Specific comments
    • Citation/Punctuation/Grammar/Syntax: Please review the manuscript for minor inconsistencies in citation, punctuation, grammar and syntax: e.g., line 136 “… GHG. [20]”, Figure 3 “Costa …”, question in line 489.

  • Introduction
    • Authors must avoid methodological description of conclusive statements of their research in the introduction section: e.g. statement related to method – “In the applied approach, the system boundaries of LCA are adjusted based on the requirements necessary for integration into ESA”; statement related to conclusion – “The advantage of the applied methodology is the capability to also account for environmental impacts other than GHG emissions, expressed as the Global Warming Potential (GWP), (e.g. resource use indicators) and the environmental impacts on inventory level (e.g. copper)”.
  • Method
    • Application: In line 102, authors state that “The basic working principle of an energy system model is that the electricity generation has to match the demand at every time step”. Nonetheless, electricity is only one form of energy. How are different energy sources and carriers considered in the proposed method?
    • Reference: The state of the art section must be underpinned by references where authors have used information or ideas from other authors or sources.
    • System boundary: In the definition of system boundaries, there is a significant interplay of trading systems in supply chains. Notwithstanding the counting of CO2-eq emissions and resource consumption only at a specific location where it taken place, it is important to acknowledge and quantify the amount of virtual resources and emissions associated with products. Consumables are only created because there is a demand for them in the market place at the end user. This latter approach hols consumers accountable/aware of their overall impact beyond the boundary of system in a more integrated systems approach.
    • Schematic flow chart: The flow chart of the methodology illustrated in Figure 3 must be defined in more details. Resources, pollutant and information flows should be better defined.
    • Contribution: The contribution of the method should be state in the conclusion in a detailed way rather than in line 285.
  • Discussion/Conclusion
    • Methodological advantages: Why the integration of LCA and ESA is critical for decision making? Why the lifecycle carbon emission reported in LCA studies cannot be used directly? Is there any modelling advantage in ESA in relation to LCA? Why the boundary of ESA cannot be expanded independently from LCA?
    • Relevance: The authors describe other studies integrating in detail LCA and ESA, including studies addressing the avoidance of double counting in carbon emissions (e.g. in line 189 authors cite Volkart [1]). In this context, what is the relevance of the proposed study for the literature? Please make it clear in the conclusion section.

Author Response

Thank you for the detailed and constructive feedback.

To improve the quality of the manuscript a full rework of the main sections was conducted, including another round of language and spell checks.

Re: General comments

The general questions you raised were addressed more precisely throughout the text and in particular in the second paragraph (decision-making) and the first paragraph of the conclusion (separate use of LCA and ESA).

Re: Specific comments

Introduction

We agree. We eliminated all methodological description of conclusive statements from the introduction.

Method

Application: The proposed methodology can also be applied to different technologies in other sectors (such as heating or transport). In this paper, the methodology is applied to the example of PERC PV and thus focuses on electricity. Additional clarifications in the introduction and conclusion were added.

Reference: The state of the art section, in particular for ESA, was reworked completely.

System boundary: Absolutely, the described method is supposed to support the correct allocation to the consuming regions (in most cases countries). This was clarified in the method section as well in the discussion.

Schematic flow chart: The flow chart in Figure 3 was updated to reflect the different flows more precisely.

Contribution: The statement in 2.2 was rewritten in a non-conclusive way and a section discussing the relevance of the method was added in the conclusion.

Discussion/Conclusion

Methodological advantages:

  • Why the integration of LCA and ESA is critical for decision making? --> Because especially when more renewable energies become part of the energy system the non-direct emission become more significant and the production becomes more and more important and thus should be considered (cf. line 505-510 and 245-249)
  • Why the lifecycle carbon emission reported in LCA studies cannot be used directly? --> As the system boundaries overlap and this would lead to double counting (cf. figure 3 and line 200-227)
  • Is there any modelling advantage in ESA in relation to LCA?
    • The advantage of modelling ESA in relation to LCA is that a more realistic representation of emission can be achieved and the problematic of carbon leakage can be addressed. This has now been explained in the introduction section (cf. line 57-61) and in conclusion (cf. line 511-517).
  • Why the boundary of ESA cannot be expanded independently from LCA?
    • This is theoretically possible from the perspective of ESA, since the system boundary of an energy system analysis is preselected based on the region that is to be analyzed. It can be on a small scale like cities and municipalities or larger scale like a country or a continent. In either way, it is unlikely that the entire value chain of all relevant components takes place in the investigated region. A consequent LCA analysis would still require the proposed adjustments.

Relevance:

The designated LCA view and the straight forward modelling process is hopefully beneficial to other LCA practitioners. This is also elaborated further in the conclusion.

Reviewer 3 Report

Please see attached

Round 2

Reviewer 3 Report

Dear authors,

Thank you for your changes and responses to my comments. I believe your manuscript is now much clearer and to the point.

I suggest you state at the beginning that you concentrate on attributional LCA only, rather than in the conclusion. I would be curious to know what changes would be needed to adapt to consequential LCA which is particularly relevant for electricity generation.

The units of the carbon intensities of per kWh could still be harmonized across the text. A final spellcheck would also be useful.

Author Response

We added clarifications that we are focusing on attributional LCA in the introduction and the state-of-the-art section.

The units for the carbon intensities were harmonized in the text and in the tables and a final spell check was conducted.

As mentioned in the funding notes the research was conducted partly in the project InteRessE. In InteRessE we are quantifying the impacts of the different energy transition pathways, in particular regarding the resource consumption in order to identify resource critical pathways. Thus, the transition pathways are determined by the ESA and we choose attributional LCA to allocate the specific impacts to the different technologies. In consequential modelling we assume that in general the issues are similar to the ones in attributional modelling (see section 2.1.3 2nd paragraph). When integrating cLCA into ESA the double counting is assumed to also be the crucial issue. In addition, and when sticking with the surrogate technology approach the double counting between the models of the different technologies in consequential models is likely to be an additional challenge. Yet, we cannot make a conclusive statement in this regard as our focus was the attributional LCA. Depending on the research aim a consequential LCA Approach might be more suitable.

Thank you again for your time and feedback.